# High-Frequency Electroporation and Chemotherapy for the Treatment of Cutaneous Malignancies: Evaluation of Early Clinical Response

**DOI:** 10.3390/cancers15123212

**Published:** 2023-06-16

**Authors:** Phoebe Lyons, Dana Polini, Kate Russell-Ryan, A. James P. Clover

**Affiliations:** 1Department of Plastic Surgery, Cork University Hospital, T12 DC4A Cork, Ireland; 2School of Medicine, University College Cork, T12 YN60 Cork, Ireland; 3Mirai Medical, H91 P5PH Oranmore, Ireland; 4Cancer Research@UCC, University College Cork, T12 YN60 Cork, Ireland

**Keywords:** high-frequency electroporation, cutaneous malignancies, electroporation, chemotherapy

## Abstract

**Simple Summary:**

Electroporation is a method of treatment in which electrical pulses are generated and applied to a cell, resulting in temporary (reversible electroporation, RE) or permanent (irreversible electroporation, IRE) changes to the structure of the cell, depending on the pulse parameters applied. There has been increasing interest in manipulating the pulse parameters used to deliver successful electroporation for cutaneous malignancies in order to reduce muscle contractions and pain associated with traditional low-frequency electrochemotherapy (LF-ECT), which often requires general anaesthesia. Higher-frequency electroporation with biphasic waves successfully permeabilises the cell membrane, usually without causing these side effects, thus allowing treatment to be delivered under local anaesthetic. This permits a wide range of patients to be safely treated, including those for which general anaesthetic is a contra-indication. This article aims to demonstrate the safety and efficacy of high-frequency electroporation, as well as equivalence and good tolerability in comparison to traditional electrochemotherapy.

**Abstract:**

High-frequency electroporation (HF-EP) with chemotherapy is a novel therapy proposed for both curative and palliative treatment of cutaneous malignancies. The use of high-frequency biphasic pulses is thought to reduce the painful muscle contractions associated with traditional electrochemotherapy (ECT), allowing treatment administration under local anaesthesia. This proof-of-concept study investigated the efficacy and tolerability of HF-EP protocols on a variety of cutaneous malignancies. A total of 97 lesions of five different histological subtypes were treated across 25 patients. At 12 weeks post-treatment, a 91.3% overall lesion response rate was observed (complete response: 79%; partial response: 12.3%), with excellent intraprocedural patient tolerability under local anaesthetic. HF-EP with chemotherapy shows promising results regarding tumour response rates for cutaneous malignancies of varying histological subtypes when compared to traditional ECT protocols. Improved patient tolerability is important, increasing the possibility of treatment delivery under local anaesthesia and potentially broadening the treatment envelope for patients with cutaneous malignancies.

## 1. Introduction

Electrochemotherapy (ECT) is a locally ablative tumour therapy which is now a well-established treatment for cutaneous malignancies and metastases, including melanoma and non-melanoma skin cancers and metastatic breast cancer [1,2]. ECT is based on the principle of electroporation, whereby a series of electric pulses are applied to cells causing the cell membrane to become transiently permeable, allowing chemotherapeutic agents to cross the cell membrane [3]. This permits chemotherapy agents with known poor cell membrane penetration, most commonly bleomycin or cisplatin administered either intravenously or intratumourally, to have a direct cytotoxic effect on tumour cells with limited effects on surrounding healthy tissue [4,5].

The incidence of all cutaneous malignancies is increasing in Ireland, particularly amongst the elderly population, creating a significant healthcare burden on the state [6]. However, due to numerous medical comorbidities or high tumour burden, many patients are unsuitable for surgical excision. Additionally, patients with cutaneous metastatic disease can be challenging to treat as a result of their debilitating associated symptoms, as well as the volume of tumour burden [7], emphasising the need for a wider range of treatment options, as well as expanding the treatment envelope. 

ECT plays a key role in treating these patient cohorts and has become increasingly more established since the development of the European Standard Operating Procedures on Electrochemotherapy (ESOPE) in 2006 and their revision in 2018 [3,8]. The efficacy of ECT as a treatment has been well demonstrated following the publication of outcome analyses collated from cancer centres across Europe, all demonstrating good response rates [1,9,10]. Furthermore, outcome analysis with regards to cutaneous malignancy histiotypes demonstrates the efficacy of ECT across multiple tumour types, including melanoma [11,12], breast cancer [13,14], and basal cell carcinomas (BCCs) [15,16], establishing ECT as a recognised treatment option contributing to the multi-disciplinary management of a variety of cutaneous malignancies.

Traditional ECT involves the delivery of low-frequency pulses and often requires general anaesthesia or sedation as a result of intra-procedural side effects including pain and muscle contractions [17]. The requirement of a general anaesthetic (GA) can make it challenging to integrate ECT into routine clinical practice. As a result, there has been increasing interest in manipulating the pulse parameters used to deliver successful electroporation (e.g., the pulse length used in high-frequency electroporation (HF-EP) is 1–2 µs [18]) in order to reduce muscle contractions and the pain associated with treatment [17], allowing the procedure to be conducted under local anaesthetic (LA). If energy was to be delivered into a 100 ohm resistor, when plugged into the equation for energy transfer E = V × I × t (where E = energy in joules (J), V = volts; I = current in amps (A); and t = time in µs) the parameters used in traditional ECT calculate an energy transfer of 1.28 J (E = 400 V × 4 A × 800 µs), whereas parameters for high-frequency electroporation equal 16.54 J (E = 525 V × 5.25 A × 6000 µs). This larger energy transfer is required as it is known that rapid bipolar pulses need larger voltage amplitudes to disrupt cells, similar to longer monopolar pulses; thus, the advantage of HF-EP with chemotherapy is that it is as efficient as the ESOPE electroporation protocol, while causing less muscle stimulation. 

Electroporation (EP) can be reversible or irreversible based on different electrical parameters including pulse duration, number of pulses, and electric field strength [19]. The electroporation window represents the necessary electric field strength versus pulse duration that is required to facilitate optimal reversible electroporation. Avoiding thermal damage and obtaining reversible electroporation is essential for the success and efficacy of ECT [20].

The inclusion of high-frequency waves and bipolar pulses in electroporation protocols has led to the development of high-frequency electroporation. The ability to perform the treatment under local anaesthesia reduces the anaesthetic risk, allowing for safer treatment on an outpatient basis [17]. Apart from the value for patients who have treatment under LA (including a decreased physical and emotional burden on the patient when the requirement for GA is removed), a significant benefit of the HF-EP approach is that it should cause less of a financial impact on the healthcare system (due to shorter procedure time, the ability to carry out treatments in an outpatient setting, a quicker turn-around time for patients with fewer post-operative beds required for observation following GA, etc.). 

Subsequently, HF-EP with chemotherapy has been validated in vitro. Scuderi et al. looked at the application of the HF-EP with a chemotherapy protocol which reduced the pulse length from 100 µs monophasic to packets of 2 µs biphasic pulses, allowing for reversible electroporation [19]. This study validated these parameters successfully in vitro on a melanoma cell line to induce cell death when combined with cisplatin while also potentially mitigating painful muscle contractions [19]. To date, there are only a small number of publications investigating reversible high-frequency electroporation and chemotherapy in the treatment of cutaneous malignancies. In one case study published out of Germany, a patient who underwent treatment of melanoma metastasis without systemic analgesia reported no interoperative or post-operative pain or muscular contractions. The patient received a total of three treatments, resulting in a significant reduction in tumour mass and associated complications [21]. In a second case study, also from Germany, a patient who was unsuitable for treatment under general anaesthetic was treated with HF-ECT under sedation and did not require post-operative analgesia [22]; the patient has undergone multiple monthly treatments with positive effect, resulting in a reduction in tumour size and as well as tumour-associated side effects such as odour and oozing [22].

The aim of this article is to demonstrate the efficacy and safety of high-frequency electroporation in conjunction with chemotherapy on various types of cutaneous malignancies. The objectives include measuring lesion response over time, in addition to showing satisfactory patient tolerability under local anaesthesia.

## 2. Materials and Methods

### 2.1. Participants

This proof-of-concept case series was undertaken at Cork University Hospital, Ireland, and includes patients that were treated with reversible HF-EP plus bleomycin between July 2019 and April 2021. For study purposes, the data cut-off for follow-up assessment was June 2022, and thus not all patients included in the dataset had been scheduled to attend for the 12- and/or 18-month follow-up visits at the time of report writing. Patients with primary cutaneous malignancies over the age of 18 were invited to enrol in the study after being deemed unsuitable for or having declined standard of care surgical management, or subsequent to the failure of initial treatment. Patients referred following discussion at multidisciplinary team (MDT) meetings for symptomatic cutaneous metastases were also invited to take part.

A total of 28 patients consented to take part in the study, of which 25 were included in data analysis; 3 patients were excluded from the analysis as they did not return for follow-up assessment. Patients that were included for data analysis ranged in age from 29 years to 93 years old and included those who may otherwise have not had a treatment option, due to contraindication to general anaesthetic. Of the 25 patients assessed, 10 patients were included for device evaluation, 14 patients were deemed unsuitable for or elected to be treated with HF-ECT due to a preference for local anaesthetic over general anaesthetic, and 4 patients experienced disease progression of their cutaneous malignancies subsequent to treatment with traditional (low-frequency) ECT. These latter patients were offered HF-EP for disease palliation as no other active alternative was available despite MDT discussion. Electroporation was achieved using an ePORE^®^ pulse generator (Mirai Medical, Galway, Ireland) and CUTIS probes (Mirai Medical, Galway, Ireland).

Clinical information collected included patient demographics, lesion type, number of lesions treated, lesion size, location of lesions, and duration of follow-up. In order to categorise lesion type, lesions were grouped based on histological diagnoses into four groups: malignant melanoma (MM), basal cell carcinomas (BCC), squamous cell carcinomas (SCC), and sarcoma and cutaneous breast metastases. 

The maximum number of lesions included for analysis per patient was seven in order to eliminate per-person response bias, and the patient’s largest lesions were included for analysis [13,19]. Categorical variables were represented by absolute numbers and percentages, while continuous variables were described by mean, standard deviation (SD), and 95% confidence intervals (CI). Ethical approval was granted from the Clinical Research Ethics Committee of the Cork Teaching Hospitals. 

The results are reported with consideration of the recommendations for reporting trials of electrochemotherapy and electroporation-based treatments [23]. 

### 2.2. Procedure

HF-EP with chemotherapy was delivered based on the previously established ESOPE protocol [3], which was developed to allow a standardisation of approach when performing traditional ECT. The only deviation from the established protocols was the pulse parameter delivered; this current standard operating protocol for electrochemotherapy uses eight 100 µs monophasic pulses of 1000 V to induce electroporation [8], as illustrated in Figure 1. 

The high-frequency protocol uses a similar total energised time of 6 ms; however, pulse delivery is in the form of 2 µs biphasic pulses in place of the monophasic pulses utilised in traditional ECT, as shown in Figure 2.

Patients received a general anaesthetic (GA), local anaesthetic (LA), or spinal anaesthetic based on patient comorbidities and characteristics, after which bleomycin was injected either intratumourally (IT) or intravenously (IV) prior to the application of the electrode. This was delivered at a dose of 1000 IU mL/cm^3^ and 15,000 IU m^2^ for IT and IV, respectively. Electroporation was achieved using an ePORE^®^ pulse generator (Mirai Medical, Galway) and CUTIS probes (Mirai Medical, Galway). The high-frequency pulse parameters are pre-determined by Mirai Medical and programmed into a high-frequency extension lead (Mirai Medical, Galway), which connects the CUTIS probe to the ePORE generator. Each high-frequency burst of energy encompasses eight individual pulses, as illustrated in Figure 3.

### 2.3. Response Evaluation

Patients returned for follow-up assessment at the following approximate timepoints: 1 month, 2 months, 3 months, 6 months, 12 months, and 18 months after treatment; however, due restrictions enforced by the COVID-19 pandemic, some patients were lost to follow-up as they were unable to attend in person for follow-up assessment. Lesion response was analysed based on the Response Evaluation Criteria in Solid Tumours (RECIST) guidelines [24]. Lesions were determined to have had a complete response (CR) with full macroscopic resolution of the tumour, or a partial response (PR) with >30% reduction in diameter of the target lesion, while disease progression (DP) was determined as a >20% increase in lesion diameter. Additionally, various lesions were labelled as unable to assess (UTA) due to the presence of lesion eschar which could generate inaccurate lesion diameters. For overall patient response, where there was a PR, DP, or UTA, the lesion with the worst response was used for overall patient evaluation.

## 3. Results

### 3.1. Patients

A total of 28 patients consented to participate, with outcome data recorded for 25 patients as 3 patients did not attend for post-treatment follow-up. Of the 25 patients included in the analysis, 9 patients completed the 18-month follow-up visit (8 of whom had complete response for all lesions treated, and 1 who displayed a complete response for one-third of lesions and a partial response for two-thirds of lesions); 5 were withdrawn due to disease progression because of a partial response of treated lesions or because the patient required retreatment for other lesions not included as part of the study; 7 patients were lost to follow-up (LTFU), of which 6 had a complete response of all lesions and one had a partial response at the last assessment they attended; and 4 patients remained on-study at the time of data cut-off. 

The mean age of participants included in the analysis was 73.6 years (SD 14.6 years). A total of 396 lesions were treated across the 25 patients, of which 97 were included for analysis (maximum of seven largest lesions per patient included), consisting of five different histological subtypes: MM (n = 51), SCC (n = 2), BCC (n = 30), sarcoma (n = 7), and cutaneous breast cancer (n = 7) (Table 1). Procedural data regarding chemotherapy administration route and anaesthesia type are represented in Table 1. All patients from both local and general anaesthetic groups reported good tolerability and muscle spasms were not an issue intra-procedurally.

### 3.2. Overall Response to HF-EP with Chemotherapy

A complete response was observed in 14 patients at three months follow-up, with a partial response seen in 5 patients, and 1 patient developed disease progression (Figure 4a). Three patients had been withdrawn/LTFU at this time point, and two patients had lesions that were unable to be assessed. Of the 22 patients that remained on study, there was an overall response (OR) rate of 86% (n = 22 patients), where overall response included partial and complete responses.

At three months post-treatment, of the 81 lesions analysed, a complete response was seen in 64 lesions and a partial response in 10 lesions, while disease progression was observed in 5 lesions (Figure 4b). Therefore, an overall response was observed in 74 lesions, representing a 91% response rate (n = 81 lesions). This is summarised in Table 2.

At the time of report writing, nine patients had completed the 18-month follow-up visit. Of the 35 lesions that were assessed at this timepoint, 33 (94%) showed a complete response (11 × BCC; 21 × melanoma; and 1 × SCC), 2 (6%) displayed a partial response (2 × BCC), and 4 patients remained on study. This is summarised in Table 3.

Figure 4 illustrates the overall patient and lesion responses at each of the follow-up visits.

### 3.3. Response According to Histological Subtype 

An observed complete response was found in 36 melanoma lesions at both three and six months (85.7% (n = 36/42)) with a partial response in six lesions (14.3% (n = 6/42)) (Figure 5a). However, disease progression was then observed in four lesions (all on the same patient) at the next follow-up visit, at the 12-month time-point, all of which had been assessed as having partially responded previously. The patient was withdrawn from analysis to undergo subsequent re-treatment with HF-ECT. 

Two patients with SCC were treated in this cohort, and both lesions showed a 100% complete response at 12 months; one patient continued to display complete response at 18 months and the other had not reached the 18-month follow-up timepoint at the time of report writing. BCCs similarly showed an adequate overall response rate (100% of lesions assessed) at the three-month time-point, with a complete response in 25 lesions (83.3% (n = 25/30)) and a partial response in four lesions (13.3% (n = 4/30)), with no disease progression visualised (Figure 5b). One BCC lesion was unable to be assessed at the three-month follow-up visit. At the 12-month time point, of the 14 patients with BCCs that were treated, 6 maintained a complete response of all lesions (12 lesions in all), 2 patients had complete response of some and partial response of other lesions (7 × CR; 3 × PR lesions), 1 had partial response of the only lesion that was treated, 4 patients were lost to follow-up (five lesions), and 1 patient who had two lesions treated remained on study.

Overall, cutaneous breast cancer lesions had the poorest response, with a complete response seen in two lesions (28.6%) and disease progression in the remaining five lesions (71.4%) at both three- and six-month follow-ups visits, at which point the patient was withdrawn from the study. Of the seven cutaneous sarcoma lesions, three had a complete response (42.9%), while four underwent a partial response (57.1%) at the one-month follow-up, after which the patient withdrew, and follow-up was not possible. However, it should be noted that there was only a single patient treated in each of these groups.

The percentage response rates included in Table 4 do not take into consideration the lesions of withdrawn patients or patients that remained on the study at the time of data analysis. 

### 3.4. Response by Drug Administration Route 

Of the 25 patients assessed, bleomycin was delivered intravenously (IV) to 12 patients, versus intratumourly (IT) to 13 patients. When broken down per lesions, however, IV was the most commonly used method of bleomycin administration, with 74.2% (72/97) of lesions treated receiving IV bleomycin and the remaining 25.8% (25/97) receiving IT bleomycin. Comparative overall response between the IV and IT administration route showed no significant difference at three months follow-up, with a complete response in 18 lesions in the IT group (72% CR (n = 18/25)) versus a complete response in 46 lesions in the IV group (63.8% CR (n = 46/72)). At the three-month follow-up, disease progression was observed in five lesions of the only patient treated for breast cancer, and who was treated with IV bleomycin. The two other lesions that were assessed for this patient showed a complete response.

Per Table 1, 12 patients were treated with IV bleomycin. At the time of the 3-month follow-up visit, two of these patients had been withdrawn, one who was treated for sarcoma (seven lesions) and another who had been treated for multiple myeloma (five lesions). The response rates in Table 5 are based on 10 patients assessed at the 3-month follow-up time point. 

Per Table 1, 13 patients were treated with IT bleomycin. At the time of the 3-month follow-up visit, one of these patients who had been treated for multiple myeloma (four lesions) had been withdrawn; thus, 12 patients and 21 lesions were available for assessment (Table 6). 

At the 12-month follow-up visit, of the 12 patients who received IV bleomycin, 5 had been withdrawn or lost to follow-up (33 lesions). Of the 39 lesions assessed, 82% (33/39) continued to show a complete response, while 8% (3/39) partially responded, and 10% (4/39) had progressed. 

Of the patients treated with IT bleomycin, three patients (nine lesions) had withdrawn or been lost to follow-up and one patient (two lesions) did not reach the follow-up timepoint; thus, 14 lesions were available for assessment. Of these, 93% (13/14; 12 × BCC; 1 × SCC) had a complete response and 7% (1/14) showed a partial response of a BCC lesion. This is summarised in Table 7. 

Of the 53 lesions assessed, an overall response rate of 93% was observed at the 12-month timepoint. 

### 3.5. Response According to Lesion Size

The majority of lesions treated were <1000 mm^3^ in size, (54.6% (53/97)), with the largest lesion treated measuring 40,810 mm^3^. Sarcoma had the largest mean size (13,098 mm^3^; SD 14,366 mm^3^), followed by BCCs (3206 mm^3^; SD 5339 mm^3^), breast cancer lesions (2380 mm^3^; SD 1873 mm^3^), SCCs (2055 mm^3^; SD 2397 mm^3^), and MMs (1010 mm^3^; SD 1777 m^3^). The complete response rate was highest in the lesions <1000 m^3^ (92.4%). However, disease progression was observed across the majority of lesion size subgroups, with the highest number lying within the 5000–5999 mm^3^ subgroup (n = 2). No clear relationship between lesion size and overall response was apparent.

## 4. Discussion

ECT is a well-established treatment for cutaneous malignancies of all histological types for both curative and palliative intent, yet painful muscle contractions remain an issue with standard protocols [1,3,7]. HF-EP with chemotherapy (ECT) is a relatively new treatment modality proposed for the management of cutaneous malignancies, of both primary and metastatic origin, that could potentially eliminate these painful muscle spasms [17,19,25]. This may allow for a broader cohort of patients to be treated as the treatment tends to be more readily tolerated under local anaesthesia. In order for HF-ECT to become commonplace in the field, the safety and efficacy of the treatment must continue to be illuminated. The first step in this process as presented here is in demonstrating the effectiveness and tolerability of this treatment across a spectrum of cutaneous histologies, thus allowing for future studies to enlarge the patient cohort. 

This proof-of-concept case series demonstrates promising results regarding the effectiveness of this HF-EP treatment protocol, with an overall response rate of 86% (complete response rate 63.6%) three months after treatment. This observed high overall response aligns with the ESOPE study published in 2006, which demonstrated an overall response of 85% (complete response rate 73.7%) [8]. A follow-on study using the InspECT database showed a similar overall response rate of 85% (complete response rate 70%, partial response 15%) [1]. In the cohort of patients treated as part of this study, the data demonstrate that HF-EP with chemotherapy is equivalent to traditional ECT with regards to overall response. These data are encouraging, especially due to the early stage inclusion criteria for this patient cohort which included patients that were deemed unsuitable for current options including those that require patients to have general anaesthetic, low-frequency ECT, and those that had progressed further to treatment with LF-ECT. The pulse parameters used in the study are pre-determined by Mirai Medical (Galway) and programmed into a high-frequency extension lead (Mirai Medical, Galway), which connects the CUTIS probe to the ePORE generator.

ECT is recognised as being efficacious across all histological subtypes [1,8]. Our study showed HF-EP was successful on MM (CR: 85.6%), SCCs (CR: 100%), and BCCs (CR: 83.3%), at the 12-week follow-up visit, yet showed reduced efficacy against sarcoma (CR: 42.9%) and cutaneous breast cancer metastases (CR: 28.6%). However, all breast cancer lesions and all sarcoma analysed were from a single patient in each instance and, as a result, may not be representative of all cutaneous breast cancer or sarcoma lesions in the field. In each case, the patients were withdrawn from the study and experienced rapidly progressive systemic disease. Despite this, the outcome data are comparable to traditional ECT and are sustained up to three months follow-up [8]. Specifically, the outcome data for melanoma (CR 85.6%) are at least comparable to data published by Kunte et al. who demonstrated an OR of 78% and a CR of 58% [12]. Similarly, CR in the BCC cohort (83.3%) was equivocal to previous studies which showed response rates of 72.5% [15], 81% after a single treatment [26], further demonstrating the efficacy of HF-EP across different histiotypes.

Certain factors are known to influence the effectiveness of traditional ECT, including lesion size, with several studies demonstrating smaller lesion sizes positively correlate with higher efficacy of ECT [1,12,27,28]. This study is too small as yet to provide a meaningful insight as to the response rates for HF electroporation with chemotherapy with regard to lesion size. There are however trends that are evident which suggest a similar correlation, which is in keeping with other published data. Thus, the poor response rates within both sarcoma and breast cancer groups, which had an average lesion size of 13,098 mm^3^ and 2380 mm^3^, respectively, may be as a result of larger lesion size, but this cannot be definitively demonstrated, and no clear relationship with lesion size and response to HF-EP with chemotherapy has been established. In this study, the overall response was highest within the <1000 mm^3^ group (92.4%). Despite this, previous studies which used traditional ECT have advocated for the early treatment of malignancies with electrochemotherapy as a result of a possible link between lesion size and treatment response [13].

Regarding route of administration, our data show that intratumoural injection has an overall comparative response to intravenous administration. Intravenous bleomycin is contra-indicated in patients with pulmonary disease due to the risk of pulmonary fibrosis [29], and intratumoural injection could prevent the potential risk for these systemic side effects and allow this cohort of patients to be treated with HF-ECT.

Traditional ECT is often associated with painful muscles spasms, and as a result, general anaesthetics are required [25], and many patients who are referred for non-operative management have significant co-morbidities that would preclude them from undergoing general anaesthesia. Approximately half the patients in this study received treatment under local anaesthetic, and each patient demonstrated excellent tolerability intra-procedurally. The ability to carry out HF-EP with chemotherapy under local anaesthetic without sedation further expands the potential of this treatment to an increasing number of patients, in particular the elderly population [30]. Overall, the average lifespan is extending, and the general population is becoming more elderly. This is resulting in an increase in the overall number of elderly patients with skin cancer [6]. With increasing age comes an inevitable increase in patient co-morbidities (e.g., heart disease, pulmonary issues) which may exclude a patient from traditional, standard-of-care treatments. As such, safe, effective, and minimally invasive treatment options for this patient group is hugely important. Electrochemotherapy has been established as a valuable treatment option for the elderly including the ‘oldest-old’ (≥90 years old) [30]. In their study carried out in 2022, Sersa et al. discovered that when compared to a significantly younger cohort of patients, the treatment outcomes were comparable between the two groups, illustrating that ECT remains efficacious in the elderly population, and maintaining efficacy and potentially diminishes the need for more complex treatments including reconstructions [30].

As mentioned previously, our small sample size is a limitation of this study, and inferences regarding the effect of different variables in response to HF-EP are limited, including lesion size, histological subtypes, and chemotherapy administration route. A possible selection bias may be present as the included patients were invited based on referrals to an ECT centre, yet within our patient cohort, patients were referred and treated for both palliative and curative intent. In addition, because a maximum of seven lesions were treated per patient, a complete picture of the patients’ disease is not evident, and despite complete response for lesions assessed as part of this study, a number of patients experienced disease progression due to systemic disease.

Limitations also exist as a result of the COVID-19 pandemic which affected the ability of patients to travel for follow-up appointments. Therefore, follow-up lesion evaluation was not always possible and some patients subsequently withdrew—these patients have been considered LTFU for data collection purposes in order to distinguish from patients with disease progression and were offered alternative treatments and withdrawn from the study. This resulted in a low number of durable follow-ups, and consequently the data are skewed towards responders. Follow-up data for those patients that were withdrawn from the study and retreated with HF-ECT has not been included in the paper, and thus the outcomes with regard to success and tolerability of repeat treatments have not been reported. An analysis of patients who had more than one treatment with HF-ECT would be beneficial, as well as publication of case reports similar to those conducted by Pfefferle et al. [21,22]. These data reported within this article, however, does establish the great potential of high frequency electroporation with chemotherapy as a treatment option. The altered pulse delivery criteria allow for good patient tolerability under local anaesthesia, and in this series displayed equivalent outcomes to standard low-frequency ECT.

Future studies should investigate the route of chemotherapeutic administration, as intratumoural administration could overcome the systemic side effects associated with intravenous chemotherapeutics including bleomycin. Ultimately, a randomised control trial with an adequately powered sample size is required to comprehensively and objectively compare traditional ECT to HF-EP with chemotherapy, in particular to assess the optimal chemotherapy administration route and response according to histological subtypes and lesion size.

## 5. Conclusions 

HF-EP with chemotherapy is a promising modality for the treatment of cutaneous malignancies, with both curative and palliative intent. This proof-of-concept study demonstrated safety of the treatment, and the efficacy of HF-EP in early clinical utility across multiple histological subtypes with good patient tolerability. This is an important step in the evolution of HF-EP with chemotherapy from traditional ECT, broadening the treatment envelope to include a larger cohort of patients with cutaneous malignancies. The field will benefit from further studies with a larger cohort of patients, as well as a greater representation of the various histological malignancies, particularly for breast cancer and sarcoma. Publication of data encompassing the safety, tolerability, and efficacy of multiple treatments with HF-ECT would also be of great value to the field.

## Figures and Tables

**Figure 1 cancers-15-03212-f001:**
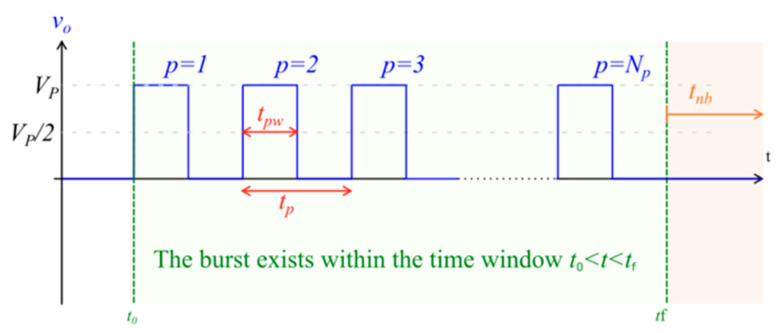
Illustration of monophasic pulse burst, which is representative of that used in traditional, low-frequency electroporation. V_p_ = setpoint pulse amplitude; N_p_ = total number of pulses in the burst; t_pw_ = pulse width (the time between the half amplitude crossings); t_p_ = pulse period; t_nb_ = time interval from the end of one burst to the start of the next burst; t_o_ = time instant at the start of the burst; t_f_ = time instant at the end of the burst.

**Figure 2 cancers-15-03212-f002:**
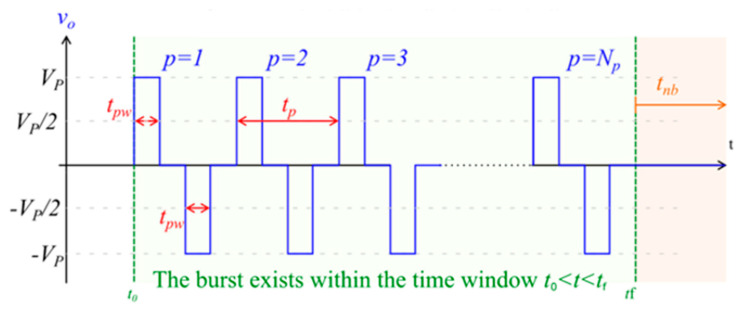
Image of biphasic pulse waveform, representative of the pulses used in high-frequency electroporation. V_p_ = setpoint pulse amplitude; N_p_ = total number of pulses in the burst; t_pw_ = pulse width (the time between the half amplitude crossings); t_p_ = pulse period; t_nb_ = time interval from the end of one burst to the start of the next burst; t_o_ = time instant at the start of the burst; t_f_ = time instant at the end of the burst.

**Figure 3 cancers-15-03212-f003:**
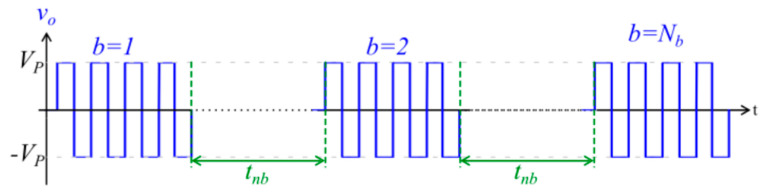
Image of high frequency burst waveform. b = burst number; N_b_ = total number of bursts; V_p_ = set voltage; t_nb_ = interburst delay.

**Figure 4 cancers-15-03212-f004:**
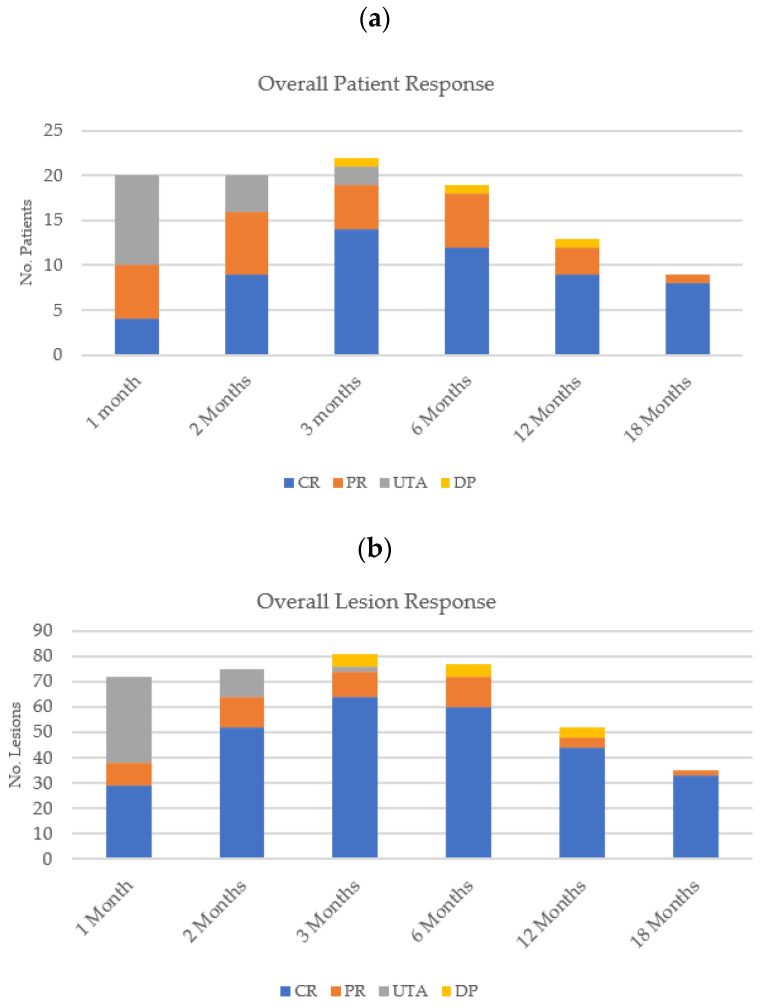
Overall (**a**) patient response and (**b**) lesion response for all histological subtypes at each follow-up time point.

**Figure 5 cancers-15-03212-f005:**
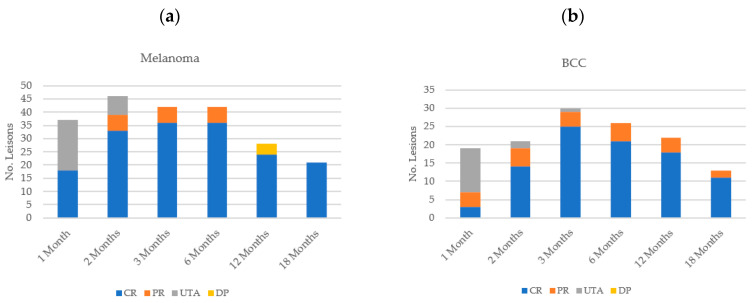
Lesion response according to histological subtype: (**a**) melanoma; (**b**) basal cell carcinoma (BCC).

**Table 1 cancers-15-03212-t001:** Patient demographics, procedural data, and breakdown of lesion types, wherein a maximum of seven lesions per patient were analysed. Refer to Section 3.4 for a breakdown of the response per route of administration.

	Mean	SD (95% CI)
Age (years)	73.6	14.6 (67.9, 79.3)
	No. pts	% of total
GA	14	56%
LA	10	48%
Spinal	1	4%
Intravenous (IV) bleomycin	12	48%
Intratumoural (IT) bleomycin	13	52%
	No. lesions	% of total
BCC	30	30.9%
SCC	2	2%
MM	51	52.5%
Breast	7	7.2%
Sarcoma	7	7.2%
Total	97	-
	Mean	SD (95% CI)
Lesion size (cm^3^)	2676.9	5742.5 (1540, 3820)

**Table 2 cancers-15-03212-t002:** Patient response rate at the 3-month follow-up visit.

Response	No. Pts	% Pts	No. Lesions	% Lesions
CR	14	63%	64	79%
PR	5	23%	10	12.30%
DP	1	5%	5	6.20%
UTA	2	9%	2	2.50%
Total	22	100%	81	100%
**OR**	**19**	**86%**	**74**	**91%**

**Table 3 cancers-15-03212-t003:** Patient response rate at the 18-month follow-up visit.

			Lesions Not Included in the 18-Month Post-Treatment Response Rate									
		Histiotype	BCC		SCC		MM		Breast		Sarcoma	
			No. of pts	No. of Lesions	No. of pts	No. of Lesions	No. of pts	No. of Lesions	No. of pts	No. of Lesions	No. of pts	No. of Lesions
No. of Pts		Withdrawn	5	12	0	0	5	30	1	7	1	7
		On Study	4	5	1	1	N/A	N/A	N/A	N/A	N/A	N/A
			Lesion Response Rate of Pts Who Completed the 18-Month Follow-Up Visit									
			BCC		SCC		MM		Breast		Sarcoma	
		No. of pts	# Lesion	% Total	# Lesion	% Total	# Lesion	% Total	# Lesion	% Total	# Lesion	% Total
Response Rate	CR	8	11	85.60%	1	100%	N/A		N/A		N/A	
	PR	1 ***	2	15.40%	0	N/A						
	DP	0	0	N/A	N/A	N/A						

*** This patient had 3 lesions assessed; 1 had a complete response and 2 had partial responses to the treatment at the 18-month follow-up visit.

**Table 4 cancers-15-03212-t004:** Histological response rate for assessed lesions at the 3-, 6-, 12-, and 18-month time points.

		3 Months		6 Months		12 Months		18 Months	
		No. of Lesions	% Total	No. of Lesions	% Total	No. of Lesions	% Total	No. of Lesions	% Total
BCC	CR	25	83.33%	21	77.80%	19	82.60%	11	85%
	PR	4	13.33%	5	18.50%	4	17.40%	2	15.00%
	DP	N/A		N/A		N/A		N/A	
	UTA	1	3.33%	1	3.70%				
	Withdrawn	N/A		3	N/A	5	N/A	12	N/A
	On study			N/A		2		5	
SCC	CR	1	50%	1	50%	2	100%	1	100%
	PR	N/A		1	50%	N/A		N/A	
	DP			N/A					
	UTA	1	50%						
	Withdrawn	N/A							
	On study							1	N/A
MM	CR	36	85.70%	36	85.70%	24	85.7	21	100%
	PR	6	14.30%	6	14.30%	0	N/A	N/A	
	DP	N/A		N/A		4	14.30%		
	Withdrawn	9	N/A	9	N/A	23	N/A	30	N/A
Breast	CR	2	28.60%	2	28.60%	N/A		N/A	
	PR	N/A		N/A					
	DP	5	71.40%	5	71.40%				
	Withdrawn	N/A		N/A		7	N/A	7	N/A

**Table 5 cancers-15-03212-t005:** A breakdown of both the per patient response rate and the histological response rate of lesions treated with IV bleomycin that were available for assessment at the 3-month follow-up visit.

	IV Bleomycin								
Histiotype	BCC		SCC		MM		Breast		Total
# Pt	2		1		6		1		10
# Lesion	10		1		42		7		60
	# Lesion	% Total	# Lesion	% Total	# Lesion	% Total	# Lesion	% Total	
CR	7	70%	1	100%	36	86%	2	29%	
PR	3	30%	0	0%	6	14%	0	0%	
DP	0	0%	0	0%	0	0%	5	71%	
**OR**	**10**	**100%**	**1**	**100%**	**42**	**100%**	**2**	**29%**	**N/A**

**Table 6 cancers-15-03212-t006:** A breakdown of both the per-patient response rate and the histological response rate of lesions treated with IT bleomycin that were available for assessment at the 3-month follow-up visit.

IT Bleomycin					
Histiotype	BCC		SCC		Total
# Pt	12 *		1 *		12
# Lesion	20		1		21
	# Lesion	% Total	# Lesion	% Total	
CR	18	90%	0	0%	
PR	1	5%	0	0%	
DP	0	0%	0	0%	
UTA	1	5%	1 **	100%	
**OR**	**19**	**91%**	**1**	**100%**	**N/A**

* One patient was treated for both a BCC and an SCC. A total of 12 patients assessed. ** Lesion UTA at the 3-month follow-up visit. A complete response was observed in this lesion at the 12-month follow up visit. At the time of writing, the patient remained on study.

**Table 7 cancers-15-03212-t007:** A comparison of the overall response rate of lesions available for assessment at the 12-month follow-up visit, based on the route of bleomycin administration.

Overall Response at the 12-Month Follow-Up Visit					
Route	IV		IT		Total
# Pt	7		7		14
# Lesion	39		14		53
On Study	0		1 patient; 2 lesions		N/A
	# Lesion	% Total	# Lesion	% Total	
CR	32 (7 × BCC; 1 × SCC; 24 × MM)	82%	13 (12 × BCC; 1 × SCC)	93%	
PR	3 (3 × BCC)	8%	1 (BCC)	7%	
DP	4 (4 × MM)	10%	0	N/A	
**OR**	**39**	**91%**	**1**	**100%**	**N/A**

## Data Availability

Research data available upon request.

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
