# Peer review of "High-Frequency Electroporation and Chemotherapy for the Treatment of Cutaneous Malignancies: Evaluation of Early Clinical Response"

_cancers, 2023, doi:10.3390/cancers15123212_

Round 1

Reviewer 1 Report

COMMENTS FOR THE AUTHORS

I found the enclosed manuscript (ID: cancers-2267594) entitled “High Frequency Electroporation and Chemotherapy for the Treatment of Cutaneous Malignancies: Evaluation of Early Clinical Response” by Lyons et al. suitable for publication in Cancers, however, minor revision should be made.

This is a presentation of results of a valuable proof-of-concept study of clinical responses to electrochemotherapy (ECT) where high frequency electroporation (HF-EP) pulses were used to deliver chemotherapeutics to cancer cells. Tumour responses to HF-EP ECT were evaluated regarding different cutaneous malignancies, drug administration route, and lesion size. The study was well conducted, however COVID pandemics led to some difficulties in a follow-up assessment. The methods should be more thoroughly described. The results and the discussion are well presented, however, a table (one or more) with more comprehensive presentation of results in addition to a text would be more informative than a present form.

Here are my comments and suggestions:

1.       Abstract: “(reversible electroporation – [RE]) or permanent (irreversible electroporation [RE])” – correct the second “RE” to “IRE”.

2.       Introduction (page 2): “whereby an electric pulse is applied to cells causing the cell membrane to become transiently permeable” – it is not always a single pulse that is enough for electroporation. In ECT, usually a series of pulses is used.

3.       Introduction (page 3): “Traditional ECT involves the delivery of low frequency pulses (pulse length 50-1000)” – unit of pulse length is missing.

4.       Methods (page 4): “this current standard operating protocol for electrochemotherapy uses eight 100 µsec monophasic pulses of 1000v to induce electroporation” – what is 1000v? 1000 V?

5.       Methods (page 4): “however the pulses are 2 µsec bihasic pulses of 1350 V/cm.” – correct “bihasic” to “biphasic”.

6.       Methods (page 4): Pulse parameters should be described more thoroughly (number, interphase and interpulse delay). An image of pulse waveform is recommended.

7.       Methods (page 4): How were pulse parameters chosen? Justification of choice of pulse parameters should be described in the introduction and/or discussion.

8.       Methods (page 4): What kind of electrodes were used in the procedure?

9.       Page 5: Define abbreviations DP and PD on the first appearance.

10.   Results (page 7): “BCCs similarly showed an adequate OR rate at three month time-point, with a CR in 25 lesions (83.3% [n=25/30]), and a PR in four lesions (13.3% [n=4/30]), with no DP visualised (Figure 2).” – here, you probably refer to the figure 3b?

11.   Results (page 7): “4 patients were lost to follow up (5 lesions) and on patient who had two lesions treated remained on study.” – correct “on patient” to “one patient”.

12.   Results should be presented in a form of a table(s) to be clearer and more informative.

Author Response

Please find attached our replies to your helpful comments in the attached word document. We feel these have helped strengthen the paper

Reviewer 2 Report

This manuscript by Lyons, et al. reviews the use of high frequency electroporation and chemotherapy for treating cutaneous lesions. It uses a fairly small sample size of 25 patients and 97 lesions. It reads fairly well but parts of it should be improved before publication. There are too many abbreviations in this paper. Some are not defined such as “DP”. I have to assume that the author meant to say “PD” for progressive disease but I should not have to guess! Please define all abbreviations when first introduced. The use of tables would greatly improve the paper as described below. Please address each of these suggestions in the revision of this manuscript.

Page       para.      line

3             2             When comparing the response to 100us pulses with 2 us biphasic, it would be good to include the energy in both pulse types. E=(V2/Z)t*N so for 8 100us pulses at 2 kV, E=32J assuming Z=100ohm. For 2us biphasic, each biphasic pulse delivers 0.16J and 200 can be delivered totalling 32J. So same energy but different muscle stimulation.

4             7             5             Please specify company and city manufacturing the ePORE as you did for CUTIS

Table 1                                  Very confusing because the intravenous Bleo is not separated from intratumoral. Please rearrange this so reader can easily see the effectiveness of each

6             1-3                         It would be much better to put this into a table listing %responses and and then describe it in text.

7             3-5                         Please put these into a table so the reader can easily see the difference between IV and IT delivery. 

Author Response

Thank you for your helpful comments and suggestions. These have been addressed in the attached word file. We feel these have strengthened the submitted paper.

Round 2

Reviewer 2 Report

This revision is much improved and is now ready for publication.